# Automatic Activity Arising in Cardiac Muscle Sleeves of the Pulmonary Vein

**DOI:** 10.3390/biom12010023

**Published:** 2021-12-24

**Authors:** Pierre Bredeloux, Come Pasqualin, Romain Bordy, Veronique Maupoil, Ian Findlay

**Affiliations:** 1EA4245, Transplantation, Immunologie et Inflammation, Groupe Physiologie des Cellules Cardiaques et Vasculaires, Université de Tours, 37200 Tours, France; come.pasqualin@univ-tours.fr (C.P.); romain.bordy@univ-tours.fr (R.B.); veronique.maupoil@univ-tours.fr (V.M.); 2Laboratoire de Pharmacologie, Faculté de Pharmacie, Université de Tours, 37200 Tours, France; ian.findlay@univ-tours.fr

**Keywords:** pulmonary veins, cardiomyocytes, automatic activity, catecholamine

## Abstract

Ectopic activity in the pulmonary vein cardiac muscle sleeves can both induce and maintain human atrial fibrillation. A central issue in any study of the pulmonary veins is their difference from the left atrial cardiac muscle. Here, we attempt to summarize the physiological phenomena underlying the occurrence of ectopic electrical activity in animal pulmonary veins. We emphasize that the activation of multiple signaling pathways influencing not only myocyte electrophysiology but also the means of excitation–contraction coupling may be required for the initiation of triggered or automatic activity. We also gather information regarding not only the large-scale structure of cardiac muscle sleeves but also recent studies suggesting that cellular heterogeneity may contribute to the generation of arrythmogenic phenomena and to the distinction between pulmonary vein and left atrial heart muscle.

## 1. Introduction

Although it has been known since the late 19th century [1], interest in the study of cardiac muscle into mammalian thoracic vein exploded with the discovery within human pulmonary veins (PV) of sites of focal ectopic electrical discharges triggering and maintaining atrial fibrillation (AF) [2].

In this context, it should be emphasized that the PV retain a venous structure, including endothelium and vascular smooth muscle layers. Exterior to these is the layer of striated cardiac muscle [3]. To our knowledge, no physical interaction (tissue abutment and/or gap junction connections) between cardiac and vascular smooth muscle in the PV has been reported. The possibility of paracrine interactions between epithelium, smooth and cardiac muscle, however, cannot be excluded.

Recently there have been special issues of both Circulation Research [4] and Cardiovascular Research [5] dedicated to the subject of AF. The interested reader should consult these for an up-to-date synthesis of the mechanisms, therapies, laboratory studies and animal models appropriate for translation to clinical situations.

In parallel with laboratory studies intended for translation to the clinic, basic physiological research has been conducted to characterize differences in the electrophysiology, pharmacology and excitation–contraction coupling of atrial muscle and that of the PV sleeves in an attempt to understand the phenomena underlying the occurrence of PV ectopic electrical activity promoting AF. It is these studies that we will consider here. We also emphasize that this represents a synthetic rather than an exhaustive review.

## 2. Pulmonary Veins: An Anatomical Substrate Favorable to the Initiation and Conduction of Ectopic Electrical Activities

### 2.1. Embryological Development of Pulmonary Veins Cardiac Muscle Sleeves

The clearest evidence for differences between cardiac muscle in the left atrium (LA) and the PV comes from studies of embryological development in the mouse [6,7]. The tracing of transcription factors Pitx2 and Tbx5 permitting the identification of entirely different origins of the PV myocardium from that of the LA. In brief, PV myocardium arises from mesenchyme tissue at the base of the PV after both the development of the LA and the establishment of the venous connection between the heart and the lung. Differentiation of this mesenchyme to the myocyte phenotype is followed over a very short period (<24 h) by proliferation and migration along the vein to intra-lobular vessels in the lung.

In the human heart, the development of PV myocardium follows similar principles [8,9]. Only after the connection of the early embryonic single PV to the LA does a myocardial sleeve develop by proliferation and migration of NKX2-5 positive and TH18 negative cells along the vein. In the 110 day human embryo, the single initial PV has divided to show four separate PV ostia draining into the LA body [8]. All four veins show cardiac muscle sleeves which at this stage extend as far as the hilum but not within the developing lung. In the adult human PV, cardiac muscle sleeves only extend 1–2 cm from the PV ostia. This is a common feature for adult PV in other large mammals, including the pig, sheep and dog. Adult rodents (mice, rats, guinea pig and rabbit) on the other hand retain PV cardiac muscle sleeves along their entire lengths including into veins in the interior of the lungs.

Taken together, these data indicate that the PV myocytes have a different embryological origin from those of the LA. It is therefore quite possible that these two populations of cells also exhibit different physiological functioning that would result in a greater propensity of PV cardiomyocytes to triggering electrical ectopic activities that promote AF.

### 2.2. The Organization of Cardiac Muscle in the Pulmonary Veins and Left Atria

The structure of human PV myocardium has been widely studied (for review see Sánchez-Quintana et al. [10]). In brief, human PV myocardial sleeves appeared to be organized as a complex mesh of muscular fibers. The LA roof between the right and left-sided PV also shows a non-uniform architecture of circular and longitudinally orientated fibers. In addition, fibrosis may create and enhance discontinuities between the myofibrils of the PV and LA roof. Similar complexity of fiber orientation has been found in the PV of the dog [11], mouse [12] and rat [3,13]. In the rat, PV consecutive cardiomyocyte layers are almost orthogonally orientated, whereas a unidirectional orientation of consecutive cardiomyocyte layers are seen in the LA [13].

At the cellular level, some studies have shown that there was no difference in size, shape or capacitance between PV and LA cardiomyocytes in dog, rabbit and rat [14,15,16]. On the other hand, other studies indicated that rat PV cardiomyocytes were larger and had a higher capacitance than those of the LA [13,17,18]. These differences might be explained by the considerable heterogeneity found within the PV cardiomyocytes population [13,18]. In particular that the incidence and organization of t-tubules in PV myocytes was highly variable from cell to cell, ranging from their virtual absence to well-organized tubular systems. These different types of myocytes were not randomly distributed in the muscle sleeves, but rather in clusters of similar cells. Clusters of PV myocytes with one type of tubular network could be surrounded by PV myocytes with another type of tubule network. The myocytes on the borders of these clusters were directly connected. All of this was in contrast to the regular transversally organized t-tubules network observed in ventricular myocytes irrespective of the species and the relative homogeneity in atrial myocytes [19]. In large mammals, the atrial cell t-tubule network is described as well-developed and extensive [20,21] whereas in small mammals the atrial cell tubular network is generally sparse and not transversally organized [22]. This cellular heterogeneity, associated with the diverse orientation of PV myocardial fibers may favor anisotropic conduction of electrical activity and facilitate reentry and the formation of rotors of ectopic foci in the PV.

### 2.3. Innervation of the Pulmonary Veins

The human PV myocardium is supplied by nerves originating from the epicardial neural plexus, formed by branches from the left and right vagus nerves, recurrent laryngeal nerves and both sympathetic trunks at the cervico-thoracic and thoracic levels [23,24,25,26]. Three epicardial neural plexi supply the four PV [26]. These ganglia contain both adrenergic and cholinergic elements [27]. Ganglionic plexi of the dog also contain both sympathetic and parasympathetic elements [28]. They also found a greater nerve density around the PV ostia where the majority of foci of ectopic electrical activity were located in humans [2], as well as a decline of nerve density from the external to the internal layer of the PV wall. The latter is consistent with human observation: nerve filaments and ganglia are located in the adventitia [29]. In the mouse, catecholamine fibers were concentrated in the PV rather than in the LA appendage [30]. In the rat, the density of catecholamine fibers in the PV increased with post-natal age [31]. These data suggest that autonomic innervation might be an important actor in the regulation of PV excitability and in the genesis of abnormal automatism triggered in the PV, as depicted in the following sections.

## 3. Pulmonary Vein Electrophysiology

### 3.1. Resting Membrane Potential in the Pulmonary Vein Myocytes

The PV cardiac muscle is reported to have a less negative resting membrane potential (RMP) than that of the LA. This was described first in the dog where it was associated with a reduced action potential (AP) amplitude and lesser maximum rate of phase 0 depolarization [11,15,32]. Relative to the LA, a lesser PV RMP has been confirmed in the rat [17,18,33,34], the mouse [35] and the guinea pig [36,37]. A consistent observation across diverse species is the decreased expression of the background inwardly rectified channel Kir2.1 in the cardiac muscle of the PV [14,16,18,37,38,39]. Other contributions to the basal conductance in at least the rat PV cardiac muscle, include an enhanced background Na^+^ permeability [17] and a hyperpolarization activated Cl^−^ current [40,41]. Neither of these were recorded in the LA. In the dog and the guinea pig, agonist independent GIRK channels could provide an additional basal K^+^ conductance [15,37]. These differences in the expression of ion channels involved in the regulation of cardiomyocyte membrane potential probably contribute to the greater arrhythmogenic potential of PV by facilitating the initiation of abnormal AP. Interestingly, in both the rat and in the guinea pig there may be a gradient of basal RMP along the length of the PV [34,42], suggesting further heterogeneity in the distribution of ion channels.

### 3.2. Conduction of Electrical Activity in the Pulmonary Veins

Differences in AP recorded from PV and LA myocardia vary across a range of species and usually involve differences in action potential duration (APD) [15,33,35,42,43]. In ex vivo preparations which included the right atrium, sinus rhythm evoked AP were recorded in the right atrium and along the PV [42,44].

While only APD differentiated the PV and LA AP in the rat under basal conditions, their reactions to selective adrenergic agonists were markedly different. The β_1_-adrenergic receptor agonist isoprenaline reduced APD in both tissues. In the PV, it also hyperpolarized plasma membrane and increased AP amplitude. The α_1_-adrenergic receptor agonist cirazoline only increased APD in the LA while it depolarized the PV myocytes membrane leading to loss of the AP and thus the tissue excitability [33]. These particular observations led us to investigate the conduction of AP in intact rat LA and PV preparations with either a linear multi-electrode array or simultaneous intracellular microelectrode recordings in the PV and LA [44]. Electrical pacing applied to the apex of the LA elicited electrical activity that propagated across the LA and along the PV. Superfusion of cirazoline led to the loss of electrical activity detectible by the electrode array in the PV. The intracellular microelectrode revealed that this resulted from depolarization of the cell membrane and the reduction of AP amplitude to a small electrotonic wave. Conduction and full overshooting AP were still visible in the LA.

In the intact rat LA and PV preparation, the gradient of depolarization along the length of the PV could sometimes lead to the loss of conduction in distal regions of the vein, which was restored by hyperpolarization evoked by acetylcholine [34]. Egorov et al. [39] found that longitudinal stretch of the PV provoked depolarization, and reduction of AP amplitude to small electrotonic waves. This was reversed by the Cl^-^ current blockers DIDS and DCPIB provoking hyperpolarization. In isolated PV myocytes, they identified I_swell_ which was blocked by DIDS and DCPIB. They also found a potential pathological role for I_swell_ in the PV of hypertensive rats [39]. In normotensive rat PV, I_swell_ was associated with Caveolin3 in the myocyte membrane. Down regulation of Caveolin3 in hypertensive rats was associated with a greatly enhanced sensitivity of PV conduction to stretch.

The evidence is therefore clear for stretch inducing depolarization in the PV. The mechanism under-laying depolarization resulting from α-adrenergic stimulation remains to be determined. In a different preparation of the rat superior vena cava, depolarization by phenylephrine was associated with reduction of the background Kir 2.1 current [45].

These recent data from the literature suggest that electrical conduction within PV is highly dependent on the activity of the autonomic nervous system and the level of stretch applied to the preparation. Furthermore, it suggests for the first time that it is possible to induce electrical disconnection of the PV by one or more pharmacological approaches, thus offering innovative research perspectives for the identification of molecular targets allowing the future development of more specific pharmacological treatments of AF by preventing the conduction of PV ectopic electrical activity to the LA.

## 4. Excitation–Contraction Coupling

In cardiomyocytes, excitation–contraction coupling is initiated by AP triggering the opening of L-type calcium channels (LCC) and calcium entry, which induces in return the release of calcium from the sarcoplasmic reticulum through ryanodine receptors (RyR). This process is known as calcium induced calcium release (CICR). Then, the Ca^2+^ released from the reticulum sarcoplasmic engage the myofilaments triggering contraction [46].

There are differences in CICR between ventricular, atrial and PV cardiomyocytes. First, L-type calcium current density is smaller in PV than in LA cardiomyocytes in rat and dog [15,18]. A T-type calcium current may also be present in PV cardiomyocytes of rabbit and rat, which may contribute to CICR [47,48].

Second, in rat the heterogeneity of cellular t-tubule organization, as described in Section 2.2, has direct consequences on the distribution of these voltage gated calcium channels which are largely expressed on t-tubules. Conversely, the spatial distribution of type 2 ryanodine receptors (RyR2) located on the sarcoplasmic reticulum (SR) membrane does not differ between ventricular, LA and PV cardiomyocytes. Thus, the number of calcium release units constituted by LCC in close vicinity to RyR2 clusters will present a great variability in PV cardiomyocytes contrary to ventricular and LA ones.

The amplitude and the spatiotemporal shape of calcium transients in PV cardiomyocytes do show considerable heterogeneity compared to those of the ventricle and atrium [13]. In dog, such differences have not been observed between PV and LA [49].

Thereby, the cellular and tissue heterogeneities may lead to non-uniformity of excitation–contraction coupling within the PV myocardial sleeves leading more easily to the generation of arrhythmogenic calcium waves, which are able to propagate within the PV. Moreover, it may also have consequences in terms of PV cardiomyocytes APD heterogeneity. Thus, localized discordance of APD and consequent refractory period could facilitate the existence of ectopic foci, mostly due to re-entry phenomena [13].

β-adrenergic stimulation substantially modifies excitation–contraction coupling and presents subtle differences between rat atria and ventricular cardiomyocytes [50]. In the PV, the consequences of cardiomyocyte heterogeneity in this context remain to be explored.

## 5. Pulmonary Veins: A Source of Spontaneous, Triggered and Catecholaminergic Automatic Activity

Spontaneous calcium events are often involved in the firing of abnormal electrical activity. Indeed, beside the singular electrophysiology of PV myocytes, some studies described intense spontaneous calcium events in rabbit and rat [13,51] whereas no difference was found in dogs [49]. In rat, calcium sparks occur more frequently in PV than in LA myocytes, which in turn leads to a higher frequency of spontaneous calcium waves. Their amplitude, duration and width were also greater in PV than in LA cardiomyocytes. Moreover, these waves were associated with depolarizing currents larger than those in LA cardiomyocytes [13]. Altogether, these results strongly suggest the involvement of calcium handling in the generation of several arrythmogenic mechanisms.

### 5.1. Spontaneous Electrical Activity

Under basal conditions, without electrical stimulation, no [33] or rare spontaneous activities [34] were observed in rat isolated PV. Some sporadic and transient depolarizations [18], but no pacemaker activity [16,41], were observed in rat isolated PV cardiomyocytes. The same has been observed in dog PV tissue preparations and isolated cardiomyocytes [11,15,52]. This absence of spontaneous activity under physiological conditions might be explained by the absence of HCN4, which underlies I_f_ in the pacemaker cells of the sinoatrial node, in either rat [53] or dog PV [15].

In guinea pig [36,37,41,42] and rabbit PV [14,41,54], spontaneous AP were observed but their incidence varied in different studies. Their frequency was lower than sinus rhythm and thus they were masked in vivo. The spontaneous AP were influenced by both cholinergic and adrenergic stimulation, as well as by stretch. Acetylcholine stopped spontaneous activity, whereas isoproterenol [14,42] and low levels of stretch (<0.3 g) [55,56] increased AP frequency. PV spontaneous AP were also blocked by the Ca^2+^ channel blocker nifedipine in the rabbit [14], and by the inhibitor of the Na^+^-Ca^2+^ exchanger (NCX) SEA0400, and ryanodine in the guinea pig [36]. Ca^2+^ release from the SR activating the forward mode of NCX could then be involved in the evocation of spontaneous AP upon the background of the reduced density of IK_1_ in the PV [14,37]. Takagi et al. [41] showed a hyperpolarization-activated inward current in guinea PV myocytes which was suppressed by Cs^+^ which block I_f_ current. This suggests the presence of this current in guinea pig PV, although to our knowledge, neither the presence of HCN4 protein and/or mRNA nor the application of the more specific I_f_ blocker ivabradine have been reported for the guinea pig PV. On the contrary, in rabbit PV there was negligible hyperpolarization-activated inward current [41] and little HCN4 mRNA [53].

In the mouse, half of the PV preparations showed spontaneous electrical activity [35]. This appeared as either constant AP firing or repetitive bursts of AP. Both were abolished by either acetylcholine or adenosine. No spontaneous activity was observed in mouse LA.

### 5.2. Triggered Activity

Because most ex vivo PV preparations from different animal species were quiescent, laboratory investigations centered upon the means to evoke arrythmogenic activity. Most of these studies involved the use of high-frequency electrical stimulation of the cardiac muscle sleeves to either induce ectopic triggered activity or to induce the classic arrythmogenic features of cardiac muscle that are early-after-depolarizations (EAD) or delayed-after-depolarizations (DAD). The hope in these investigations was to demonstrate the character or simply the incidence of these phenomena, which could be specific to PV rather than a general condition of the LA myocardium.

In human and dog PV, a protocol of high-frequency pacing, a pause and then a single stimulus would result in tachycardia. In the human PV, this post-pause activation could arise from a sinus beat [57]. In the dog PV the post-pause single stimulus was not effective under control conditions, but required the superfusion of norepinephrine (NE) or isoprotenerol and acetylcholine to provoke tachycardia [57,58]. Ryanodine abolished both EAD formation during the sympathetic stimulation and triggered bursts during the combination of sympathetic and parasympathetic stimulation [58].

In what could be clearly a species difference, a post-pause stimulus train applied to the rabbit PV actually required ryanodine to evoke sustained triggered activity. This effect was enhanced by isoproterenol and reduced by the calcium depletion of the sarcoplasmic reticulum with cyclopiazonic acid [59].

In the guinea pig, ryanodine blocked pacing triggered extra systoles in the PV [60]. In the rat, atrial remodeling and especially increases in tissue fibrosis are pre-requisites for pacing to induce either tachycardia or fibrillation [61,62,63,64].

In a series of experiments conducted upon the canine PV, Patterson et al. [52] combined direct stimulation of cardiac muscle to evoke AP in the vein with high-frequency short duration stimulation to the ganglionic plexus innervating the vein. The coincidence of ganglionic stimulation with an evoked AP resulted in the reduction of AP duration and the triggering of tachycardia. As for pacing induced tachycardia [58], the effects of ganglionic stimulation were blocked by atropine, atenolol and ryanodine [52]. Human ganglionic plexus ablation has gone on to prove an adjunct to PV ablation in the clinic [65,66,67]. It also suggests an involvement of both sympathetic and parasympathetic nervous systems in abnormal electrical activity.

### 5.3. Catecholaminergic Automatic Activity

Ex vivo rat PV responded to NE with bursts of spontaneous contractions. This catecholaminergic automatic activity (CAA) which was independent of electrical stimulation, required the simultaneous activation of α_1_ and β_1_ adrenergic receptors and was not observed in the LA [68]. Intracellular microelectrode recording showed that the response of the PV quiescent membrane potential to NE was biphasic. First, a hyperpolarization resulting from activation of β_1_ adrenergic receptors was observed. It was followed by depolarization resulting from the activation of α_1_ adrenergic receptors that preceded the onset of CAA. Automatic activity presented as bursts of slow rising phase and low amplitude AP. Moreover, automatic AP showed variable form and frequency during these bursts. Intervals were characterized by hyperpolarization at the end of the burst and slow depolarization to the onset of the next. The effects of NE upon the quiescent membrane potential in the LA were slight and without evocation of CAA [33]. Doisne et al. [33] also showed that upon a continuous background of α-adrenergic receptors stimulation, different doses of β-adrenergic receptors agonist evoked different forms of automatic activity (Figure 1). These ranged from activities with varying burst and inter-burst intervals to activity reduced to isolated low-frequency AP arising from a relatively depolarized membrane potential. The highest dose of isoprenaline actually abolished automatic activity. Figure 2 illustrates that different forms of automatic activity in the rat PV can result from slight changes in the concentration of NE. Rather than blocking CAA, tetrodotoxin evoked changes by reducing burst duration and increasing interval length [17].

In an intact PV-LA preparation of the rat, Egorov et al. [34] also showed that NE evoked a biphasic response in the PV membrane potential, which could result in automatic activity. This automatic activity presented as continuous firing in the majority of cases. They revealed complex interactions between automatic AP arising from separate sites in the vein and with electrical pacing of the LA [34]. The reactions of the rat PV to adrenaline were also dose dependent [69]. Low adrenaline concentrations induced depolarization and loss of electrical conduction, whereas high concentrations led to hyperpolarization, restoration of intra-PV conduction and the induction of automatic AP activity. Moreover, moderate stretch applied to the PV facilitated the development of arrythmogenic activity induced by a high concentration of adrenaline [69]. The predisposition of rat PV to show CAA evoked by NA has been found to develop with age and the post-natal development of the sympathetic innervation [31].

These data may contribute, at least in part, to the explanation of the increased incidence of AF observed with age and in patients with certain chronic cardiovascular pathologies, such as hypertension and heart failure.

In isolated rat PV cardiomyocytes, spontaneous and sustained AP induced by NE were recorded with the perforated patch clamp technique [18]. In these cells, NE induced automaticity was blocked by SEA0400, a selective NCX inhibitor, as well as by 2-APB, a functional antagonist of IP_3_R [18], suggesting a functional coupling between the NCX and IP_3_R in the t-tubule micro-domain. These data were then incorporated into a mathematical model of the PV cardiomyocyte [70]. However, in another study, NE failed to induce automatic activity in rat isolated PV cardiomyocytes [16], perhaps because of differences between the patch clamp methods.

In quiescent mouse PV [30,35], NE also induced automatic activity that appeared in the different forms (continuous or bursts of AP) already observed in spontaneously active preparations under basal conditions (Section 5.1). NE was without effect on the LA. However, Potekhina et al. [30] showed that automatic activity could also be evoked by isoprenaline or phenylephrine alone, which seems to be clearly a species-specific feature. Both substances caused hyperpolarization of the PV membrane potential. Acetylcholine and adenosine inhibited NE-induced activity in the mouse PV [35].

NE was also shown to induce a CAA in the distal end of quiescent guinea pig PV preparations [42]. There was a gradual transition in the form of the spontaneous AP, from sharp early repolarizations at the distal end to atrium like AP in the proximal PV. This NE-induced automatic activity in guinea pig PV depended, as in the rat [33,68], on both α_1_ and β_1_-adrenergic receptors activation [71].

A question that follows from the induction of automatic activity in muscle sleeves of the PV is that in order for such activity to evoke and/or sustain AF, it has to be conducted to the atrium.

CAA in the rat PV [33,34,69] is characterized by slow AP [33] that are not blocked by doses of tetrodotoxin sufficient to abolish AP in the LA [17]. The combination of these elements suggests that they represent slow Ca^2+^ current rather than fast Na^+^ current induced AP which characterizes electrical activity of non-pacemaker cardiac muscle. Therefore, it is likely that conduction of these slow AP is limited by the space constant. The results of Egorov et al. [34,69] showed some examples of low amplitude automatic AP evoked by either NE or adrenaline in the distal vein resulting in clear sharp over-shooting AP in the PV ostium. No details have been provided regarding the rates of AP rising phase. Although in rats, the mapping of foci in the vein and their success or failure to conduct and convert to atrial AP remains to be determined; mapping in the mouse showed that foci located in the PV ostium did show transmission to the LA [30].

### 5.4. Other Thoracic Veins

Although this review focuses on the physiology of the PV myocardial sleeves, similar results have also been observed in the rat superior vena cava (SVC) and azygos vein [31,45,72], in which myocardial sleeves also extend and constitute a source of arrhythmia in humans [2]. The effects of age on the induction of CAA in the SVC and PV was also explored [31]. CAA occurred mainly in the SVC of young animals and its incidence decreased with age to become rare in adult animals. On the other hand, CAA was rare in the PV of young animals and its incidence increased with age to become dominant in the adult.

## 6. Conclusions

The mechanisms involved in the onset of arrhythmia within PV have been widely studied. In particular those that lead to the development of abnormal action potentials but which are not specific to the PV. Indeed, PV cardiac muscle electrophysiological and calcium handling properties as well as innervation make it particularly suitable for triggered activity.

However, in addition to these well-known arrythmogenic mechanisms, the PV structure and singular cellular physiological properties open the door for other ways to trigger arrhythmia. CAA illustrates one arrythmogenic mechanism specific to the PV. The fine mechanics of CAA induction remains to be determined but requires further investigation of the cell physiology of PV cardiomyocytes and especially their regulation by adrenergic pathways. To which can be added, the singularity of fiber orientation and the heterogeneity of the cell population which is specific to PV myocardium contributing to its arrythmogenic potential.

Finally, recent data from basic research suggest that it is possible to induce electrical disconnection of the PV from the LA by one or more pharmacological approaches. This offers, for the first time, innovative research perspectives for the identification of molecular targets allowing the future development of more specific pharmacological treatments of AF by preventing the conduction of PV ectopic electrical activity to the LA.

## Figures and Tables

**Figure 1 biomolecules-12-00023-f001:**
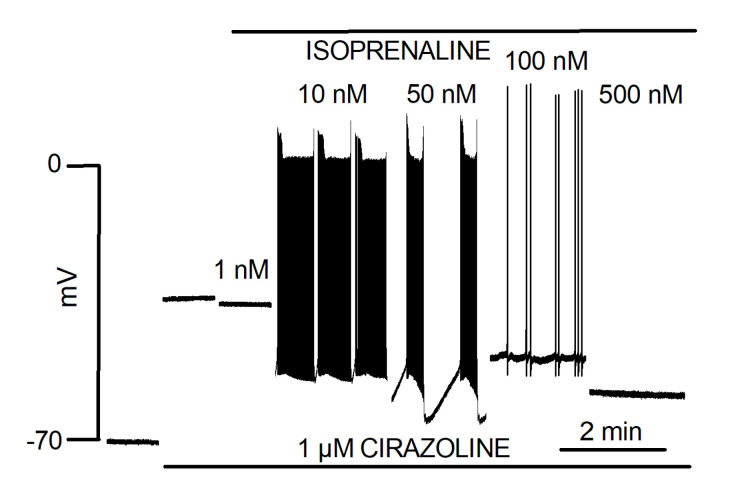
Automatic activity in cardiac muscle of the PV produced by the combination of α- and β-adrenergic receptor stimulation. Traces represent isolated segments of an otherwise continuous recording of membrane potential in cardiac muscle of one PV during the superfusion of 1 µM Cirazoline and different concentrations of isoprenaline. Modified from Doisne et al. [33].

**Figure 2 biomolecules-12-00023-f002:**
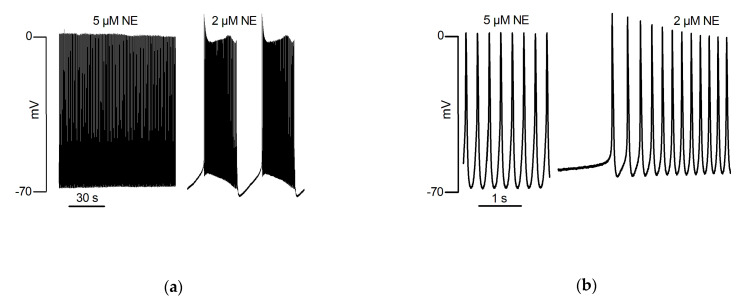
An intracellular microelectrode recording from an isolated PV of the rat. These traces form parts of an otherwise continuous recording during one penetration. (**a**) Automatic activity which had been evoked by the superfusion of 5 µM NE presented as continuous firing (3.8 Hz) of action potentials arising from a diastolic membrane potential of −69 mV, overshooting to +4 mV with a maximum rate of phase 0 depolarization of 5.4 V/s. Ten minutes after reducing the concentration of NE to 2 µM, continuous firing was replaced by automatic activity representing as bursts separated by silent intervals. Details of individual automatic action potentials recorded under these conditions are shown upon an expanded scale in (**b**).

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
