# Peer review of "Automatic Activity Arising in Cardiac Muscle Sleeves of the Pulmonary Vein"

_biomolecules, 2021, doi:10.3390/biom12010023_

Round 1
Reviewer 1 Report
In this paper, the electrophysiological and pharmacological properties of pulmonary vein myocardium, which play important roles in the induction and maintenance of atrial fibrillation, were compared with those of left atrial muscle. The factors responsible for the generation of spontaneous and evoked electrical activity were summarized. Each information was organized by animal species and described in detail, from tissue structure to innervation and electrophysiological properties, providing helpful information to understand arrhythmias originating in the pulmonary vein myocardium.
For each item, detailed information was organized by animal species and presented, but I think there is little description of the physiological or pathophysiological significance of the information. As a result, it gives the impression of just listing information, and it isn’t easy to understand what the paper is trying to achieve as a whole.
Author Response
We agree. As we were getting together the information for this short review we were struck by the lack of cross-species mechanisms, at least in the small mammals. Therefore, we felt that it would be important to at least bring this information together. Our conclusions, we hope are clear, was then to outline that cellular heterogenicity might form a common feature that requires further investigation.
We have now further clarified the objectives of this review in the introduction (line 45):
“In parallel with laboratory studies intended for translation to the clinic, basic physiological research has been conducted to characterize differences in the electrophysiology, pharmacology and excitation-contraction coupling of atrial muscle and that of the PV sleeves in an attempt to understand the phenomena underlying the occurrence of PV ectopic electrical activity promoting AF.”
We have added more descriptions regarding the physiological/pathophysiological significance of key informations:
- In the section 2.1 : Embryological development of pulmonary veins cardiac muscle sleeves (line 72):
“Taken together, these data indicate that the PV myocytes have a different embryological origin from those of the LA. It is therefore quite possible that these two populations of cells also exhibit different physiological functioning that would result in a greater propensity of PV cardiomyocytes to triggering electrical ectopic activities that promote AF.”
- in the section 2.3 : Innervation of the pulmonary veins (line 113 and 118):
“They also found a greater nerve density around the PV ostia where the majority of foci of ectopic electrical activity were located in human [2]”
“These data suggest that autonomic innervation might be an important actor in the regulation of PV excitability and in the genesis of abnormal automatism triggered in the PV as depicted in the following sections.”
- in the section 3.1 : Resting membrane potential in the pulmonary vein myocytes (line 133):
“These differences in the expression of ion channels involved in the regulation of cardiomyocyte membrane potential probably contribute to the greater arrhythmogenic potential of PV by facilitating the initiation of abnormal AP.”
- in the section 3.2 : Conduction of electrical activity in the pulmonary veins (line 172):
“These recent data from the literature suggest that electrical conduction within PV is highly dependent on the activity of the autonomic nervous system and the level of stretch applied to the preparation. Furthermore, it suggest for the first time that it is possible to induce electrical disconnection of the PV by one or more pharmacological approaches, thus offering innovative research perspectives for the identification of molecular targets allowing the future development of more specific pharmacological treatments of AF by preventing the conduction of PV ectopic electrical activity to the LA.”
- in the section 4 : Excitation-contraction coupling (line 199):
“Thereby, the cellular and tissue heterogeneities might lead to non-uniformity of excitation-contraction coupling within the PV myocardial sleeves leading more easily to the generation of arrhythmogenic calcium waves able to propagate within the PV. Moreover it may also have consequences in terms of PV cardiomyocytes APD heterogeneity. Thus localized discordance of APD, and consequent refractory period could facilitate the existence of ectopic foci, mostly due to re-entry phenomena [11].”
- in the section 5.3 : Catecholaminergic automatic activity (line 318):
“These data may contribute, at least in part, to the explanation of the increased incidence of AF observed with age and in patients with certain chronic cardiovascular pathologies such as hypertension and heart failure.”
We have also completed the conclusion (line 386):
“Finally, recent data from basic research suggest that it is possible to induce electrical disconnection of the PV from the LA by one or more pharmacological approaches. This offers, for the first time, innovative research perspectives for the identification of molecular targets allowing the future development of more specific pharmacological treatments of AF by preventing the conduction of PV ectopic electrical activity to the LA.”
We hope that these modifications will have clarified the text.
Reviewer 2 Report
Comments To the authors (1520209):
The authors summarize the ectopic activity in the pulmonary vein cardiac muscle, which could induce human atrial fibrillation. Thus, the authors attempted to summarize the physiological phenomena underlying the occurrence of automatic activity in animal pulmonary veins. It is also emphasized that cellular heterogeneity could significantly contribute to the generation of arrythmogenic phenomena and to the distinction between the pulmonary vein and left atrial heart muscle.
The review is well written and the summary is also correct. Thus, the review manuscript is a good and short summary of several decades regarding to the arrhythmogenesis in the heart. Indeed, many factors play important roles in the genesis of arrhythmias, including both in the right atrium and left ventricle under various conditions of ischemia and reperfusion via the modification of the shape of action potential (AP). These include, e.g., the role of pulmonary vein, vein insufficiency, excitation contraction coupling, modification of sodium/calcium currents and exchanges, reactive oxygen radicals, catecholamines, triggered activities and so on. All of the aforementioned pathogenic components result in several changes in the shape of the cardiac AP and functions of various ion channels, leading to the development of severe arrhythmias and syndromes, which result in the development of sudden cardiac death under both experimental and clinical conditions.
This reviewer believes that this short review manuscript is a very valuable on, however, some additional suggestions (please, see below) are recommended as the followings, and thus, an additional paragraph should be added in the revised version, before accepting for publication:
The right atrial and left ventricular arrhythmias are frequently causing sudden cardiac death, and the pathological stage of pulmonary veins (PV), e.g., vein insufficiency, can significantly contribute to the development of life threatening right atrial and left ventricular arrhythmias by several pathological factors and mediators, as some of them have been mentioned above.
Therefore, at the end of the manuscript (before the “Conclusions”, part 6., line 346), this reviewer suggests an additional paragraph, mentioning several molecules, transmitters, and factors, which can significantly contribute to the development of right atrial and left ventricular arrhythmias leading to sudden cardiac death. Thus, in this context, some valuable and so-called ‘classic publications’ should be discussed, acknowledged and cited in the revised version, as the followings:
- J Mol Cell Cardiol. 1978 Jan;10(1):81-94. doi: 10.1016/0022-2828(78)90008-1
- J Mol Cell Cardiol, 1987 Sep;19(9):841-51.doi: 10.1016/s0022-2828(87)80613-2
- J Mol Cell Cardiol. 1988 Mar;20(3):181-5. doi: 10.1016/s0022-2828(88)80051-8
- Am Heart J. 1990 Oct;120(4):819-30. doi: 10.1016/0002-8703(90)90197-6
- Cardiovasc Drugs Ther. 1991 Feb;5(1):191-200. doi: 10.1007/BF03029820
- Arrhythm Electrophysiol Rev. 2015 May;4(1):9-13. doi: 10.15420/aer.2015.4.1.9. Epub 2015 Mar 15
- J Gen Physiol. 2019 Sep 2;151(9):1066-1069. doi: 10.1085/jgp.201912409. Epub 2019 Aug 20
- Front Pharmacol. 2020 May 12;11:616. doi: 10.3389/fphar.2020.00616. eCollection 2020
- Biophys J. 2021 Jan 19;120(2):352-369. doi: 10.1016/j.bpj.2020.12.001. Epub 2020 Dec 15
- 2021 Feb;77(2):605-616. doi: 10.1161/HYPERTENSIONAHA.120.14858. Epub 2020 Dec 28
- Interface Focus. 2021 Feb 6;11(1):20190124. doi: 10.1098/rsfs.2019.0124
Finally, in summary, the aforementioned publications, as valuable and classic ones, are recommended to be acknowledged and cited as an additional chapter (before the “Conclusions”, part 6., line 346) in the revised version of this manuscript. The incorporation of these aforementioned and valuable papers into the revised version may significantly increase the interest between basic scientists and clinicians, connecting as a bridge between clinical and basic sciences.
C
Therefore, at the end of the manuscript (before the “Conclusions”, part 6., line 346), this reviewer suggests an additional paragraph, mentioning several molecules, transmitters, and factors, which can significantly contribute to the development of right atrial and left ventricular arrhythmias leading to sudden cardiac death. Thus, in this context, some valuable and so-called ‘classic publications’ should be discussed, acknowledged and cited in the revised version, as the followings:
- J Mol Cell Cardiol. 1978 Jan;10(1):81-94. doi: 10.1016/0022-2828(78)90008-1
- J Mol Cell Cardiol, 1987 Sep;19(9):841-51.doi: 10.1016/s0022-2828(87)80613-2
- J Mol Cell Cardiol. 1988 Mar;20(3):181-5. doi: 10.1016/s0022-2828(88)80051-8
- Am Heart J. 1990;120(4):819-30. doi: 10.1016/0002-8703(90)90197-6
- Cardiovasc Drugs Ther. 1991 Feb;5(1):191-200. doi: 10.1007/BF03029820
- Arrhythm Electrophysiol Rev. 2015 May;4(1):9-13. doi: 10.15420/aer.2015.4.1.9. Epub 2015 Mar 15
- J Gen Physiol. 2019 Sep 2;151(9):1066-1069. doi: 10.1085/jgp.201912409. Epub 2019 Aug 20
- Front Pharmacol. 2020 May 12;11:616. doi: 10.3389/fphar.2020.00616. eCollection 2020
- Biophys J. 2021 Jan 19;120(2):352-369. doi: 10.1016/j.bpj.2020.12.001. Epub 2020 Dec 15
- 2021 Feb;77(2):605-616. doi: 10.1161/HYPERTENSIONAHA.120.14858. Epub 2020 Dec 28
- Interface Focus. 2021 Feb 6;11(1):20190124. doi: 10.1098/rsfs.2019.0124
Author Response
We sincerely thank the second reviewer for his comments and suggestions to improve this manuscript.
We agree that the role of cellular heterogenicity in ventricular myocardium in sudden death and ventricular fibrillation is now established but at the present time to link this phenomenon to emerging data from the pulmonary vein and left atrium might be considered to be premature.
Indeed, the myocardium of the pulmonary veins is the main source of ectopic electrical activity that triggers episodes of paroxysmal atrial fibrillation (pAF), the first stage of the disease. pAF is a rhythm disorder that primarily affects the left atrium. Although it can have repercussions on the electrical activity of the ventricles and serious long-term complications (stroke, heart failure), it is not, unlike ventricular fibrillation and to our knowledge, directly responsible for sudden death.
Rather, that cellular heterogeneity as a general contribution to cardiac arrythmia is a subject that would benefit from an extensive review of its own.
To try to answer reviewer 2, we have now further clarified the objectives of this review in the introduction.
Finally, this review article, if accepted, would be part of a special issue entitled "molecular pathogenesis of cardiac arrhythmia". We believe that among the other articles that will constitute this issue, some will probably already deal with elements that are of particular interest to the reviewer 2.